# DEEP NEAREST CLASS MEAN CLASSIFIERS

**Samantha Guerriero**
VANDAL Laboratory
Sapienza Rome University

**Barbara Caputo**
VANDAL Laboratory
Italian Institute of Technology

**Thomas Mensink**[*]
Computer Vision Group
University of Amsterdam

## ABSTRACT

In this paper we introduce DeepNCM, a Nearest Class Mean classification method enhanced to directly learn highly non-linear deep (visual) representations of the data. To overcome the computational expensive process of recomputing the class means after every update of the representation, we opt for approximating the class means with an online estimate. Moreover, to allow the class means to follow closely the drifting representation we compare online-mean updates to mean condensation and decay mean updates. Using online class means DeepNCM can train efficiently on large datasets. Our experimental results on CIFAR-10 and CIFAR-100 indicate that DeepNCM performs on par with SoftMax optimised networks.

## 1 INTRODUCTION

We are interested in Nearest Class Mean (NCM) classifiers, a well-known model for multi-class classification, where a test image is classified to the class with the closest class data mean. This in contrast to the SoftMax classifier, where for each class a weight vector is learned, in NCM the class representation is based solely on the mean representation of the images belonging to that class. While NCM has gained popularity and has widely been used, *e.g.* in the Fisher Discriminant Analysis and in the Rocchio classifier [Webb (2002); Manning et al. (2008)], it has often been stated to be weak because it is a linear classifier, and recently due to its inability to learn deep representations.

In this paper, we introduce our ongoing work on DeepNCM, which learns directly a highly non-linear deep representation while enforcing the desired property that images will be mapped closer to their class mean than any other class means. DeepNCM can be considered as the extension of the Metric Learning approach for NCM (Mensink et al., 2013), where a *shared visual representation* has been learned on top of hand designed fixed features, which generalises well over novel classes and therefore could be successfully used for class incremental learning.

We believe that DeepNCM, to date, has not yet been fully explored because computing class means over a changing representation would be overly expensive. Here, we circumvent expensive updates by approximating class means with an online estimate. This approximates the real mean, while guaranteeing high adaptability to the changing deep representation as new samples are processed.

**Relation to other approaches**    Most notably, our proposed DeepNCMs relate to:

(i) The incremental classifier learning approach of Rebuffi et al. (2017), where NCM classifiers are applied on SoftMax learned visual representations. While NCM classifiers are used, they do use a representation learned using a (rather) standard SoftMax classifiers, on a fixed-size training size to meet memory constraints, and with a distillation loss to overcome catastrophic forgetting. Our proposed DeepNCM would allow to directly learn the visual representation.

(ii) The few-shot learning approach of Prototypical Networks (Snell et al., 2017), where a deep representation is learned based on NCM classifiers, however they consider the scenario of episode-few-shot classification (Vinyals et al., 2016), where each episode is a classification task using just a handful classes and data points. Computing exact class mean over all images in an episode is in this scenario trivial. In contrast to DeepNCM, their method would not extend to a larger number of images or classes, *e.g.* when not all classes are present in a (mini-)batch.

---

[*]thomas.mensink@uva.nl

## 2 DEEP NEAREST MEAN CLASSIFICATION

The nearest class mean (NCM) classifier is a distance-based classifier, which assigns an image to the class with the closest mean:

$$y^\star = \operatorname*{argmin}_{y \in \{1,...,Y\}} d(\boldsymbol{x}, \boldsymbol{\mu}_y), \tag{1}$$

where $Y$ denotes the number of classes, class mean $\boldsymbol{\mu}_y = \frac{1}{N_y} \sum_{i:y_i=y} \boldsymbol{x}_i$, and $N_y$ the number of examples in class $y$. The success of NCM classifiers critically depends on the distance metric used.

Mensink et al. (2013) proposed a probabilistic interpretation of the class mean distances, to learn a (squared) low-rank Mahalanobis distance, by maximising the log-likelihood:

$$\mathcal{L} = \frac{1}{N} \sum_i^N \ln p(y_i|\boldsymbol{x}_i), \qquad p(y|\boldsymbol{x}) \propto \exp -\tfrac{1}{2} d_{xy}^W, \qquad d_{xy}^W = (\boldsymbol{x}-\boldsymbol{\mu}_y)^\top W^\top W(\boldsymbol{x}-\boldsymbol{\mu}_y) \tag{2}$$

where $\boldsymbol{x} \in \mathbb{R}^D$ and $W \in \mathbb{R}^{m \times D}$, with $m \leq D$ as intrinsic dimension of the metric space. The final NCM classifier can be identified as a multi-class SoftMax classifier, with constrained bias terms and weight vectors, e.g. the weight vector is constraint to: $\boldsymbol{w}_y = W^\top W \boldsymbol{\mu}_y$, see (Mensink et al., 2013).

**NCM for Learned Representations** Instead of relying on a fixed representation and learning a Mahalanobis metric $W$, the current standard is to learn the parameters of a deep representation $\phi(\cdot)$. We postulate that the highly non-linear nature of deep representations eliminates the need of a linear metric $W$ and allows to use the Euclidean distance between the deep representations:

$$d_{xy}^\phi = (\boldsymbol{\phi}(x) - \boldsymbol{\mu}_y^\phi)^\top (\boldsymbol{\phi}(x) - \boldsymbol{\mu}_y^\phi), \qquad\qquad \boldsymbol{\mu}_y^\phi = \frac{1}{N_y} \sum_{i:y_i=y} \phi(x_i), \tag{3}$$

where $\phi(x)$ is the deep representation of image $x$, which evolves during training. For example $\phi(x)$ could be the penultimate layer of a ConvNet, such as AlexNet or ResNet.

**Mean Updates** It is – in theory – straightforward to learn $\phi(\cdot)$ via the gradients of the log-loss by stochastic gradient descent. However, due to the drift caused by learning the representation, after each mini batch of images, (re-)computing class means would cost a full epoch over the train data. This makes training prohibitively (computationally) expensive for large datasets.

Approximating the class means with the per-batch class means is not a sound solution and has two main disadvantages. First, when using small batch sizes (e.g. due to a large network structure) the batch class mean is likely to be a poor approximation of the real mean. Second, when not all classes are present in a batch, ad hoc modifications to the classification problem are required, i.e. removing classes not present in a given batch will change the classification problem and will change the normaliser of the probability distribution per batch. This latter will become more severe when small batch sizes are used, especially in combination with a large number of classes in the dataset.

### 2.1 MEAN UPDATES PER BATCH

To circumvent expensive mean updates, we propose to use an online estimate of the class means:

$$\boldsymbol{\mu}_{y_i}^\phi \leftarrow \frac{n_{y_i}}{n_{y_i}+1} \boldsymbol{\mu}_{y_i}^\phi + \frac{1}{n_{y_i}+1} \phi(x_i), \tag{4}$$

where $n_{y_i}$ denotes the current number of samples in class $y_i$. The class means are updated with the relevant samples every batch, following smoothly the drifting learned representation $\phi(\cdot)$.

We consider also two alternatives:

**Mean Condenstation** When computing the online mean, the most recent example counts as much as the first example, while the representation has drifted since. To assign more weight to recent examples, we introduce per epoch mean condensation. At the start of each epoch, we set $n_y = 1 \,\forall y \in Y$, i.e. the *historical* class means count just as a single data point in the current epoch.

**Decay Mean** In stead of resetting the mean after each epoch, we use a decay scheme:

$$\boldsymbol{\mu}_{y_i}^\phi \leftarrow \alpha \, \boldsymbol{\mu}_{y_i}^\phi + (1 - \alpha) \, \phi(x_i), \tag{5}$$

where in practice we use the batch mean $\boldsymbol{\mu}_{By}^\phi$, instead of a single example $\phi(x_i)$.

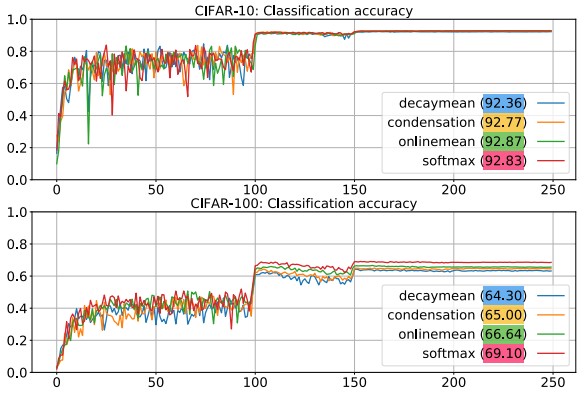

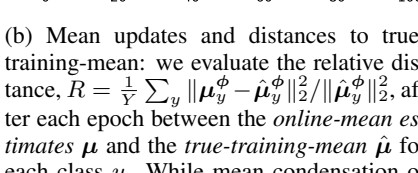

(b) Mean updates and distances to true-training-mean: we evaluate the relative distance, $R = \frac{1}{Y} \sum_y \|\boldsymbol{\mu}_y^{\boldsymbol{\phi}} - \hat{\boldsymbol{\mu}}_y^{\boldsymbol{\phi}}\|_2^2 / \|\hat{\boldsymbol{\mu}}_y^{\boldsymbol{\phi}}\|_2^2$, after each epoch between the *online-mean estimates* $\boldsymbol{\mu}$ and the *true-training-mean* $\hat{\boldsymbol{\mu}}$ for each class $y$. While mean condensation or decay decreases the relative mean distances, this is not reflected in accuracy gain.

(a) Test Top-1 performance evaluation over train epochs

Figure 1: Experiments on CIFAR10 and CIFAR100 datasets. When using ResNet-32 representations, all our proposed DeepNCM variants are on par with the state-of-the-art SoftMax baseline.

## 3   EXPERIMENTS

In this section we describe some of our initial experiments using the proposed DeepNCM classifier on the MNIST, CIFAR-10 and CIFAR-100 datasets, reporting Top-1 accuracy. All models are trained using SGD with Momentum, with dropout, explicit weight regularisation and gradient clipping $(-1, 1)$, using the learning-rate schedule devised for softmax-based ResNets[1].

**Comparison to SoftMax**   First, we compare the NCM classifiers to SoftMax classifiers, the current standard for image classification, on multi-class image classification. We have evaluated a fixed pixel-based representation and a shallow ConvNet on MNIST and CIFAR-10, where DeepNCM was on par with the SoftMax-based classifier (results not included). This encouraged us to continue to learn representation based on a ResNet-32 architecture (He et al., 2015). From the ResNet-32 network point of view, the NCM classifiers uses the penultimate layer as $\phi(x)$ and the class means are based on this penultimate layer, the network is trained from scratch.

The results for CIFAR-10 and CIFAR-100 for different variants are shown in Fig. 1a. Where the decay rate ($\alpha = 0.9$), the condensation rate (1 epoch), and learning rate (1e-01) are based on a first set of experiments on CIFAR-10, both the decay rate and the condensation rate seem to have little impact on performance (on CIFAR-10). From the results we conclude that DeepNCM is able to learn strong visual representations, and performs on par with the SoftMax based classifier.

**Relative Mean Distance**   Here we compare the relative mean distance between the online mean estimate and the true-training-mean of the DeepNCM variants. The hypothesis is that mean condensation and decay mean have smaller distances, since they focus more on the newest examples, and thus adjust faster towards the learning/moving representation. The results in Fig. 1b confirm this hypothesis, and show that indeed the relative mean distance is smaller. However, this does not (directly) returns in a higher classification rate (see Fig. 1a). A possible reason is that the internal parameters of the momentum optimiser, should be adjusted for the changing learning objective (*e.g.* after the condensation step the first few epochs could cause large changes in the mean).

## 4   CONCLUSION

We have introduced DeepNCM, an NCM based deep learning framework, which is computationally feasible due to the use of online class mean approximations, training is improved by using mean condensations. Experimentally we have shown that DeepNCM offers a compelling alternative to training (standard) SoftMax-based ConvNets. Our current work is in extending DeepNCM in the context of class incremental learning and open-set classification, where class means seem a natural way to describe categories, to incorporate new classes, and to define class boundaries.

---

[1]We follow the TensorFlow Models ResNet implementation available at github.com/tensorflow/models

ACKNOWLEDGMENTS

This research was supported by the ERC grant 637076 - RoboExNovo and the NWO VENI What & Where project. Code for DeepNCM is available on GitHub (github.com/tmensink/deepncm).

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
