# OpenReview forum: "DeepNCM: Deep Nearest Class Mean Classifiers"
_ICLR.cc/2018/Workshop — Accept_

### Official Review · AnonReviewer1 · 2018-03-12
**DeepNCM: Deep Nearest Class Mean Classifiers**

**Rating:** 5
**Confidence:** 3

**Review:**

The paper proposes an enhanced version of a Deep version of the Nearest Class Mean classification method (DeepNCM). This enhanced version is based on using the data in each training mini-batch to efficiently update the class means. In other words, they use an online approximation for the class means. This type of approximation of the means of different groups has been used before in several techniques, as an example, incremental clustering using K-means. In this sense, the proposed contribution is marginal and I do not see a relevant impact.

In terms of results, authors show that DeepNCM performs similarly to Soft-Max in small data image classification cases such as MNIST, CIFAR-10, and CIFAR-100, although, soft-max still outperforms DeepNCM in this cases. In this sense, testing is limited and still  Soft-Max offers better performance.

---

### Official Review · AnonReviewer3 · 2018-03-16
**I like the NCM idea in general but some key baselines should be tried**

**Rating:** 7
**Confidence:** 4

**Review:**

This paper proposes an extension of the nearest mean classifier by Mensink et al. that applies a deep neural network to the underlying features, rather than computing NCM in the original data space under a learned Malahanobis metric. To deal with the issue of scaling to large datasets, they propose two techniques: online mean updates and mean condensation. Online mean updates simply refers to using an online update formula to calculate the mean. This is technically incorrect though, as the underlying representation changes with each step, and further, each new point’s contribution to the current mean gets smaller with time. To counteract this, mean condensation simply resets the count of the number of examples to 1 at each epoch.

The experiments show that mean condensation helps to stabilize deep NCM training, that deep NCM performs slightly comparably (albeit slightly worse) than softmax, and that NCM provides a good representation for learning new classes.

The results are still preliminary and don’t quite demonstrate the benefits of an NCM classifier over a standard softmax, but this is a workshop and I do think that NCM-like ideas are worth exploring. I would recommend a baseline of using an exponential moving average to compute the online mean updates, as opposed to the current online arithmetic average. It would take this form:

\mu <- \beta * \mu + (1 - \beta) * mean(\phi(x))

Where mean(\phi(x)) is the mean over the current minibatch. This would allow you to backpropagate through some of the mean computation, and avoid the issue of a stale mean. I would set \beta to something like 0.9 here.

An alternative baseline is to directly parameterize \mu as weight vectors like a softmax, but compute the softmax energy as -||\phi(x) - \mu_i||^2 (for class i). This is a hybrid that might still allow you to approximate a new class using the average of \phi(x). It’s also equivalent to a constrained softmax.

These baselines should help to settle whether online updates + mean condensation is actually necessary.

I’d like to see a version of Figure 1, where the curves indicate train/test error of online NCM vs online + mean condensation. Also, how do these compare to softmax? Do they learn faster or slower?

For a conference version of the paper, I definitely recommend exploring some of the settings suggested in the conclusion.

Questions:
With online mean updates, are you backpropagating through the computation of \mu at all? Otherwise wouldn’t the gradient be biased (incorrect) with respect to the objective?
What does SFT-50 in table 1b refer to? Did you train a softmax classifier and use the representation with NCM + fine-tuning? I think this result warrants more explanation.

---

> ### Author Response · Authors · 2018-04-26
> **Thank you for the suggestions, we have incorporated them.**
>
> Dear reviewer,
>
> Thank you for the extensive feedback. We have included your suggestions and updated the paper (and experiments) accordingly. However, this took quite some time, therefore I can only now respond on your review.
>
> In the new version, we have removed the generalisation experiments, to make place for more extensive online mean, mean condensation and mean decay experiments. We show learning curves and the relative mean distance. (And more experiments are in the Github online). From the results we conclude the following: while decay and condensation have a positive impact on the relative mean distance, their performance is (slightly) lower than normal online means. Albeit all similar to softmax baseline.
>
> The idea of "learning" the weights is also interesting. We're aiming to include that in a future version.

---

### Decision · Program_Chairs · 2018-03-20
**ICLR 2018 Workshop Acceptance Decision**

**Decision:**

Accept

**Comment:**

Congratulations, your paper was accepted to the ICLR workshop.

---

> ### Public Comment · ~Thomas_Mensink1 · 2018-04-26
> **Thanks! See you next week.**
>
> Thanks! See you next week.